# A novel, benchtop model for quantitative analysis of resistance in ventricular catheters

**Pranav Gopalakrishnan**[1], **Ahmad Faryami**[2], **Carolyn A. Harris**[3]*

**1** Department of Medical Education, Wayne State University School of Medicine, Detroit, MI, United States of America, **2** Department of Biomedical Engineering, Wayne State University, Detroit, MI, United States of America, **3** Department of Chemical Engineering and Materials Science, Wayne State University, Detroit, MI, United States of America

* caharris@wayne.edu

## Abstract

### Introduction

The mechanisms of catheter obstruction are still poorly understood, but the literature suggests that resistance to fluid flow plays a significant role. We developed and assessed a gravity-driven device that measures flow through ventricular catheters. We used this device to quantitatively analyze the resistances of unused ventricular catheters used in the treatment of hydrocephalus; failed hydrocephalus catheters from our catheter biorepository were also evaluated quantitatively.

### Methods

Catheters of three manufacturing companies were inserted into the benchtop model, which records time, flow rate, and pressure data using sensors. The relative resistances of catheters across six design models were evaluated. Experiments were performed to evaluate changes in the relative resistance of a catheter when the catheter's holes were progressively closed. The relative resistance of explanted catheters from our catheter biorepository was also measured.

### Results

Experimental results showed significant differences ($P<0.05$) between the relative resistances of different catheter models just after being removed from their packaging. A non-linear trend of increasing resistance was observed in experiments on catheters with artificially obstructed holes. Data from five individual benchtop models were compared, and the differences in measured data between the models were found to be negligible. A significant increase ($P < 0.05$) in relative resistance was observed in explanted catheters.

### Conclusion

The current study sought to propose a novel in-vitro model and use it to examine data on differences in relative resistance among catheter models. From these experiments, we

**Data Availability Statement:** All relevant data are within the paper.

**Funding:** Research reported in this publication was supported by the National Institute of Neurological Disorders and Stroke of the National Institutes of Health under award number R01NS094570. Approximately 60% of this project was financed with federal dollars. The content is solely the responsibility of the authors and does not necessarily represent the official views of the National Institutes of Health. The funders had no role in study design, data collection and analysis, decision to publish, or preparation of the manuscript.

**Competing interests:** The authors have declared that no competing interests exist.

**Abbreviations:** CSF, Cerebrospinal Fluid; CNS, Central Nervous System; CFD, Computational Flow Dynamics; WSU, Wayne State University; ICP, Intracranial Pressure; DLP, Digital Light Processing; VCTD, Ventricular Catheter Testing Device; M1M1, Manufacturer 1 Model 1; M1M2, Manufacturer 1 Model 2; M2M1, Manufacturer 2 Model 1; M2M2, Manufacturer 2 Model 2; M3M1, Manufacturer 3 Model 1; M3M2, Manufacturer 3 Model 2; 0RO, Zero Rows Obstructed; 1RO, One Row Obstructed; 2RO, Two Rows Obstructed; 3RO, Three Rows Obstructed; 4RO, Four Rows Obstructed; ANOVA, Analysis of Variances.

can rapidly correlate clinical patient cohorts to identify mechanisms of luminal shunt obstruction.

## Introduction

Hydrocephalus is a chronic disorder causing abnormal enlargement of the cerebral ventricles. The most common treatment for hydrocephalus is using surgically implanted shunts to drain excess CSF to other body parts, such as the peritoneal cavity. However, these shunts have notoriously high failure rates: approximately 50% of pediatric hydrocephalus shunts fail within two years of implantation, and virtually all hydrocephalus patients will require at least one shunt revision during their lifetime [1, 2].

Despite the incredible quantity of work on the subject, the mechanisms of shunt obstruction are still poorly understood. Many studies have investigated how factors such as flow dynamics, hole geometry, and changes in resistance impact obstruction [3, 4]. To facilitate further research on this topic, a multicenter national biorepository was created at Wayne State University (WSU). This biobank was used in a study to examine the effects of numerous factors on the number of revisions undergone by patients and for imaging analysis on failed ventricular catheters [5, 6]. The literature suggests that catheter geometry, obstruction, and resistance to CSF flow are interlinked factors of shunt failure. Thus, it would be useful to have a device that researchers can use to collect and analyze substantial amounts of quantitative flow data from catheters used to treat hydrocephalus.

## Materials and methods

### The design of the benchtop testing device

This model used a water column to simulate a range of hydrostatic pressure, similar to changes in intracranial pressure (ICP). The model, hereafter referred to as the Ventricular Catheter Testing Device (VCTD), was a simple in-and-out system with a position open for an in-line shunt catheter (Fig 1). Water from the column flowed through the catheter before draining out through the silicone outlet tubing. The model held a maximum of 10 mL of water. This volume of water was chosen to minimize the risk of damage to biobank samples while still providing a sufficient pressure drop for analysis. The VCTD was comprised of three fused deposition modeling 3D-printed components (Fig 1A). The sample catheters were placed into a chamber made of UV-sensitive resin. A digital light processing (DLP) 3D printer was used to manufacture all components.

An AMD ELEGOO UNO R3 Arduino-compatible board was connected to the VCTD for data measurement and storage (Fig 1D). The VCTD measures pressure in units of hPa and fluid flow in microliters/min. The VCTD was designed for automated data acquisition: data were automatically obtained from pressure sensors (Adafruit MPRLS Ported Pressure Sensor, 0–25 PSI) and flow sensors (Sensirion AG SLF3S-1300F, 40 mL/min) positioned distal to the sample holder. The pressure of the water column was also measured using a pressure sensor at the bottom of the column, proximal to the sample holder. The proximal pressure sensor also measured the atmospheric pressure to account for differences in testing site altitude.

### Procedure for testing catheters using the VCTD

The procedure for using the VCTD was as follows: first, the catheter was cut to a length of 40 millimeters (starting from the tip) using the groove on the side of the VCTD. Next, the tubing

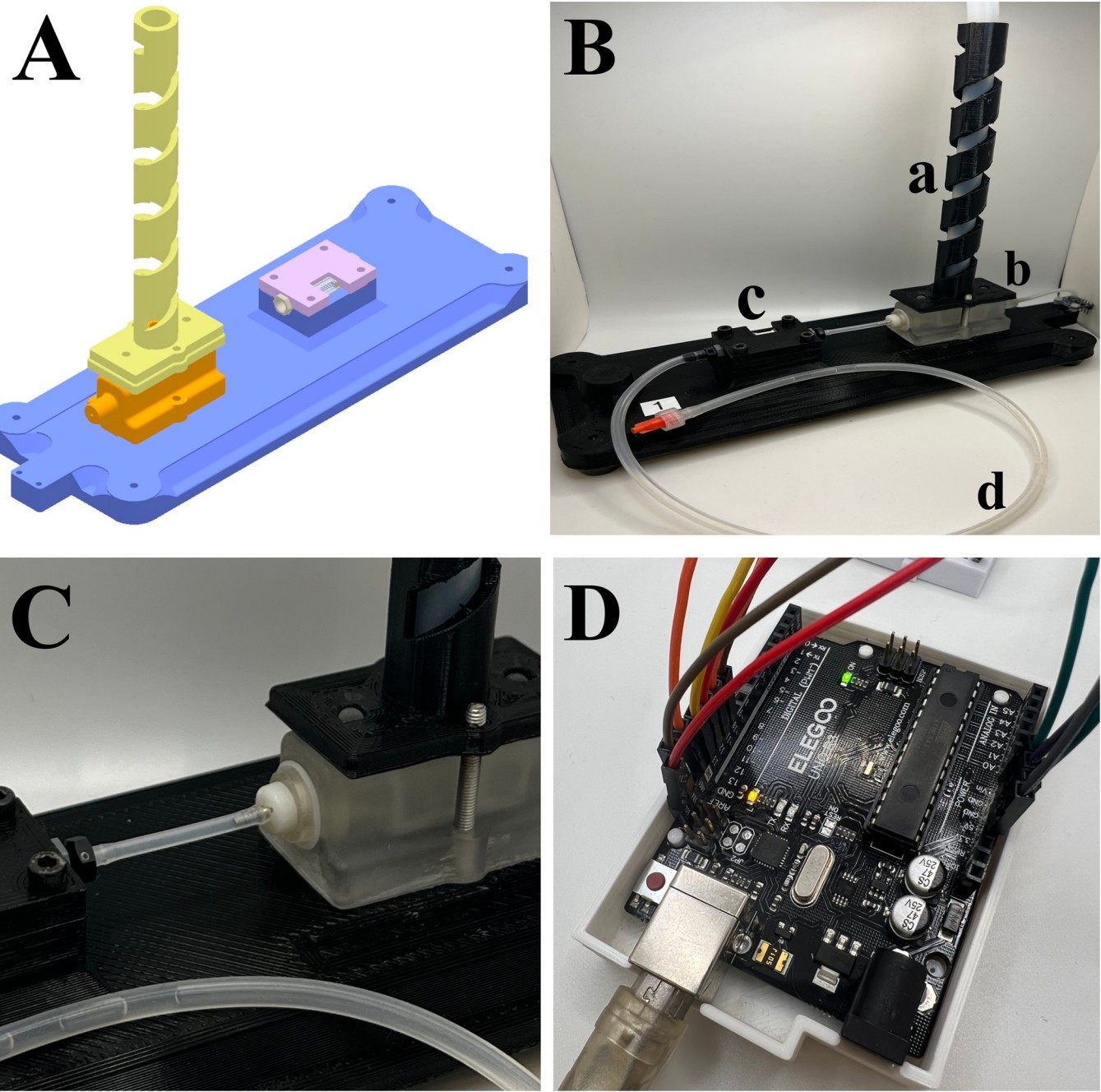

**Fig 1. The ventricular catheter testing device.** A is a CAD model of the VCTD. The VCTD consists of the water column (yellow), the sample chamber (orange), and the sensor chamber (lavender), all attached to a single-piece base (blue). B is an image of one 3D-printed VCTD. The major components of the VTCD are shown in B: the water column (a), sample chamber (b), sensor chamber (c), and silicone outlet tubing (d). C is the sample chamber. After removing the silicone tubing, the sample holder can be removed from the chamber by turning the holder counterclockwise. D is the AMD circuit board. The two lights visible are the yellow light and green light that notify the operator of the VCTD's operational status.

that connects the flow sensor to the sample holder was removed. The sample holder (Fig 1C) was unscrewed and detached from the device. The sample catheter was eased onto the sample holder, ensuring that the sample was securely and entirely on the sample holder. Next, the sample holder was screwed back into the VCTD, and the previously removed tubing was

reattached. Fluid flow was shut off using the clamp to pin down the end of the tubing. The column was filled with water up to the top using a syringe. Once the column was filled, the device was plugged into an outlet. Before continuing, the operator waited for the green light to turn on and for the yellow light to turn off (Fig 1D).

Once the device was ready to collect data, the plug was removed to allow fluid to flow. The VCTD was not disturbed while the water was flowing. The operator waited five seconds after the flow ceased before unplugging the VCTD to stop data acquisition. After testing, the sample catheter was removed, and the sample holder was wiped using a Kimwipe and 60% w/v ethanol. No particular containment procedure was needed to store the VCTD, but it was ensured that the electronic components avoided contact with fluids. After use, the VCTD was allowed to air dry. Distilled, deionized water was used for all experiments and for rinsing the VCTD's tubing.

### Experimental hypotheses and methods

Two hypotheses were evaluated: (1) unused ventricular catheters of different models will have significantly different resistances, and (2) resistance will increase as more catheter holes are obstructed. Data from individual 3D-printed devices, without any catheters inserted, were also assessed to determine whether differences in data measurement between devices were significant.

We performed four categories of experiments: (1) experiments testing five individual VCTD devices with no catheters inserted, (2) experiments with unused catheters inserted into the VCTD, (3) experiments with patient-explanted catheters inserted into the VCTD, and (4) experiments with an progressively obstructed catheter inserted into the VCTD. All experiments with catheters were performed using VCTD #1 (n = 5 for all experiments).

We created a catalog of commercial catheters to help select catheters for testing. Unused, unexpired catheters were acquired directly from manufacturers (or via eSutures). Two different models from three different biomedical manufacturing companies were tested, totaling six different ventricular hydrocephalus catheter models. Explanted hydrocephalus catheters were obtained from Wayne State University's biobank of consented neurosurgical samples. All samples presented in this study were given generic labels (Table 1).

The experiments measuring flow through the VCTD without any inserted catheter were performed on five different VCTD models (n = 5). One unused M1M1 catheter was obstructed

**Table 1. Identifiers and basic dimensions of unused and explanted catheters presented in this study.**

| Identifier | # of Rows | # of Holes per Row | Inner Diameter (mm) | Outer Diameter (mm) |
|---|---|---|---|---|
| M1M1 | 4 | 8 | 1.3 | 2.5 |
| M1M2 | 4 | 8 | 1.3 | 2.5 |
| M2M1 | 4 | 10 | 1.3 | 2.5 |
| M2M2 | 4 | 10 | 1.27 | 2.54 |
| M3M1 | 4 | 5 | 1.3 | 2.5 |
| M3M2 | 4 | 5 | 1.5 | 3.0 |
| Explanted 1 | 4 | 8 | NA | NA |
| Explanted 2 | 4 | 8 | NA | NA |
| Explanted 3 | 4 | 8 | NA | NA |
| Explanted 4 | 4 | 5 | NA | NA |
| Explanted 5 | 4 | 8 | NA | NA |

The inner and outer diameters for the explanted catheters were unknown as the model of those catheters was not determinable at that time.

artificially in the lab using transparent, Gorilla-brand hot glue sticks. A hot glue gun was used to apply a thin layer of glue to each row of holes. Care was taken to ensure the hot glue gun's tip did not contact the catheter. The catheter was not heated appreciably during the application of glue. Firm pressure was applied to the glue-coated surface while the glue was cooling to ensure the plugging of the holes. Once the glue had fully solidified, the catheter was placed inside the VCTD for testing. Data for the artificially obstructed catheter were categorized based on how many rows of holes were plugged with glue. For example, data for the unobstructed catheter were labeled as 0RO (i.e., Zero Rows Obstructed).

## Data curation and analysis

Once pressure, flow, and time data were acquired, the hydrostatic pressure resulting from the water column at each data point was calculated by subtracting the end pressure (i.e., atmospheric pressure) from the total pressure at each data point. A range of pressures, from 15 hPa to 7 hPa, was selected to ignore noise observed at the beginning and end of each experiment (Fig 2A). The chosen range used approximately 600 data points.

Once the hydrostatic pressure was found for the dataset, the time elapsed from 15 hPa to 7 hPa was calculated. This calculation was performed for each run, and then the average time elapsed (in seconds) along with standard deviations for each experimental group. The average time elapsed and standard deviation data were used for statistical analysis.

Time elapsed was used to compare differences in resistances between groups. The relationship between pressure, flow rate, and resistance for a Newtonian fluid through a tube is well-known: resistance is a function of differential pressure over bulk flow rate (Eq 1) [7, 8].

$$R = \frac{\Delta P}{q} \tag{1}$$

The pressure drop for each experiment was approximately the same; hence, differences in flow rate between catheters resulted mainly from differences in resistance. A longer time elapsed meant a greater resistance to flow, and a shorter time meant a lesser resistance to flow. The time elapsed for a fixed pressure drop was termed "relative resistance" and used as a proxy measurement for resistance to CSF flow.

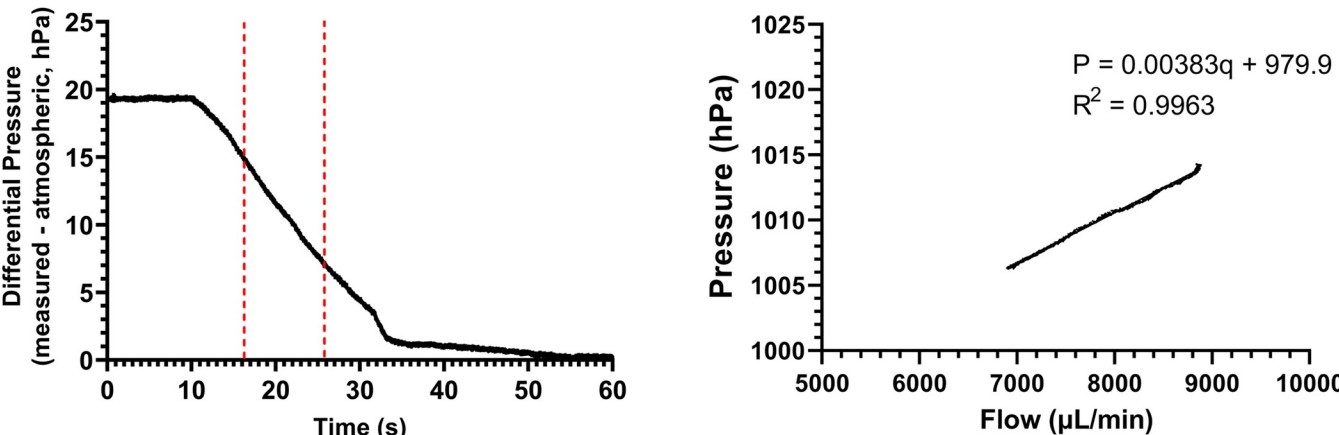

**Fig 2. Representative examples of graphs.** A shows a representative pressure vs time graph constructed using data from an unused M1M1 catheter. Data is from one experimental run of an unused hydrocephalus catheter of model M1M1. The section of data enclosed by the red, vertical lines was used for data analysis. The enclosed data region represents the time elapsed for hydrostatic pressure to drop from 15 hPa to 7 hPa. B shows a representative example of a pressure vs flow graph created using data from the same M1M1 catheter. A rolling average of 110 data points was used for the flow data when constructing the graph; the rolling average was used to reduce noise from data measurement.

To determine the validity of relative resistance as a metric for catheter comparison, catheter resistances were calculated using pressure and flow rate data. An excellent study by Cheatle et al. calculated resistances for catheters used to treat hydrocephalus [9]. Using VCTD-measured data, pressure vs. flow graphs were constructed for each dataset (Fig 2B) to calculate resistances. Linear regression was performed on each dataset using these graphs. The relationship between pressure, flow, and resistance was used to determine that the slope of a given sample's pressure vs. flow graph was the resistance for that sample. The slope and $R^2$ values for each linear regression were calculated and recorded. A dimensional analysis was performed to convert from units of hPa•min/μL to Pa•s/m³ to match units with the literature (Eq 2). Calculated resistance values were compared to the resistance values reported in the literature.

$$R\left(\frac{\text{Pa} \cdot \text{s}}{m^3}\right) = R\left(\frac{\text{hPa} \cdot \min}{\mu\text{L}}\right) \cdot \frac{60\left(\frac{s}{min}\right) \cdot 10^9\left(\frac{\mu\text{L}}{m^3}\right)}{0.01\left(\frac{hPa}{Pa}\right)} \tag{2}$$

## Statistical analysis

Microsoft Excel for Windows was used to collate data. Statistical analyses were performed using SPSS v28 for Windows and an alpha value of 0.05. Time elapsed data were analyzed for normality by constructing histograms. A one-way analysis of variances (ANOVA) test was used to determine significant differences between data of a given experimental group; a Tukey post-hoc analysis followed each ANOVA. Box-and-whisker plots were constructed using SPSS; all other graphs were made using GraphPad Prism version 9 for Windows. Data from separate hypothesis categories were not compared to each other. For example, data from the unused catheter with progressively plugged holes were not compared to data from the other unused catheters.

## Results

### Artificially obstructed catheter

The results for the artificially obstructed catheter (Table 2) showed that the differences in time elapsed were significant ($P < 0.001$). Post-hoc analysis (Fig 3A) further showed that the time

**Table 2. Time elapsed data and resistance calculations for all groups presented in study (with standard deviations).**

| Identifier | N | Mean Time Elapsed ± Standard Deviation (s) | Time Elapsed Range (s) | Resistance (Pa•s/m³, $10^{10}$) | Resistance Std. Dev. (Pa•s/m³, $10^9$) |
|---|---|---|---|---|---|
| 0RO | 5 | 11.9789 ± 0.37433 | 0.93 | 1.10 | 1.49 |
| 1RO | 5 | 12.0512 ± 0.38948 | 1.01 | 1.02 | 0.774 |
| 2RO | 5 | 12.4762 ± 0.55951 | 1.37 | 1.04 | 0.388 |
| 3RO | 5 | 13.3693 ± 0.59252 | 1.28 | 1.10 | 0.628 |
| 4RO | 5 | 25.6485 ± 1.16637 | 2.94 | 2.04 | 0.636 |
| M1M1 | 5 | 9.9852 ± 0.54147 | 1.41 | 2.29 | 2.07 |
| M1M2 | 5 | 9.3130 ± 0.32093 | 0.79 | 1.26 | 0.422 |
| M2M1 | 5 | 9.1012 ± 0.57967 | 1.48 | 1.30 | 0.725 |
| M2M2 | 5 | 9.5582 ± 0.68280 | 1.64 | 1.36 | 1.21 |
| M3M1 | 5 | 9.3614 ± 0.44481 | 1.10 | 1.23 | 2.53 |
| M3M2 | 5 | 8.6576 ± 0.43962 | 0.89 | 1.19 | 1.53 |
| VCTD 1 | 5 | 8.5761 ± 0.07193 | 0.18 | 2.28 | 1.56 |
| VCTD 2 | 5 | 7.9875 ± 0.23304 | 0.53 | 1.23 | 1.38 |
| VCTD 3 | 5 | 9.1629 ± 0.34848 | 0.94 | 1.24 | 0.354 |
| VCTD 4 | 5 | 7.4064 ± 0.21453 | 0.57 | 1.39 | 1.32 |
| VCTD 5 | 5 | 7.7334 ± 0.17477 | 0.44 | 1.31 | 1.87 |

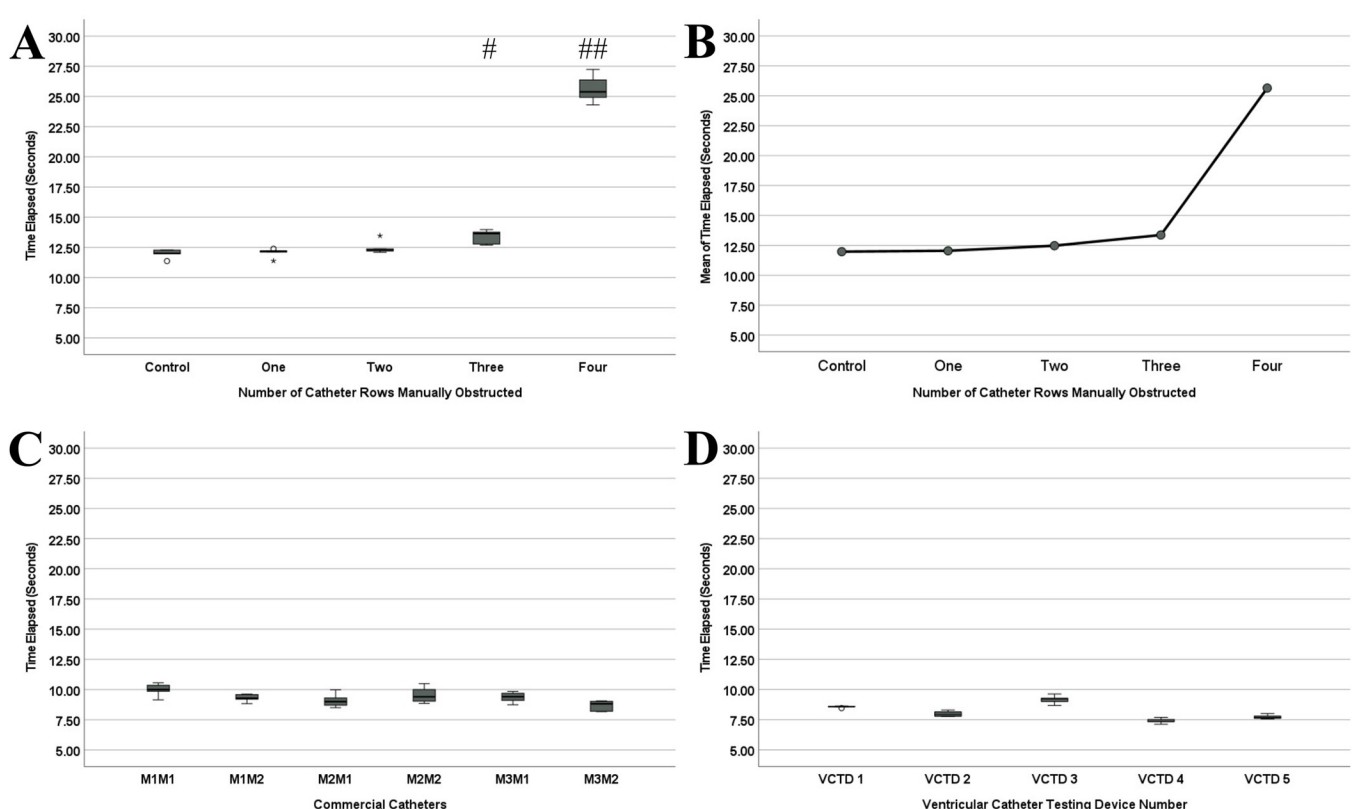

**Fig 3. Collage of graphed experimental results.** A is the box-and-whiskers plot of the mean time elapsed data for the artificially obstructed catheter. The pound symbols represent significant differences from all other groups. B shows the trend in relative resistance between the means of the groups for the artificially obstructed catheter experiments. C is the box-and-whiskers plot of the mean time elapsed data for the unused commercial catheters. D is the box-and-whiskers plot of the mean time elapsed data for the individual VCTDs.

elapsed for the four-rows-plugged catheter was significantly different from the time elapsed for all other groups ($P < 0.001$). Furthermore, the time elapsed for the unplugged catheter and one-row-plugged catheter were significantly different compared to the time elapsed for the three-row-plugged catheter ($P < 0.05$). The time elapsed data for the unplugged catheter, one-row-plugged catheter, and two-rows-plugged were not significantly different from each other ($P > 0.05$). Similarly, the time elapsed data for the two-rows-plugged and three-rows-plugged catheters were not significantly different ($P > 0.05$). A trend of increasing relative resistance was observed as holes were plugged; the trend appears to be exponential (Fig 3B).

## Commercial catheter comparison

The results of the experiments on commercial catheters (Table 2) showed that there were significant differences between the time-elapsed data for different models ($P < 0.05$). Post-hoc analysis (Fig 3C) showed that the differences between time elapsed data for only two catheter models, M1M1 and M3M2, were significantly different ($P = 0.005$). Conversely, the time elapsed data between all other model pairs was insignificant ($P > 0.05$).

## Comparison of data measurement among individual VCTDs

The results (Table 2) found that the differences between the time elapsed data for each VCTD were significant ($P < 0.001$). The post-hoc analysis (Fig 3D) showed that VCTD #1 was

significantly different from all other VCTDs ($P < 0.005$). Furthermore, VCTD #2 was significantly different from VCTD #3 ($P < 0.001$) and VCTD #4 ($P = 0.005$). Finally, VCTD #3 was significantly different than VCTD #4 ($P < 0.001$). VCTD #2 and VCTD #5 were not significantly different ($P > 0.05$). The data for VCTD #4 and VCTD #5 were also not significantly different ($P > 0.05$).

## Calculation of catheter resistance using pressure and flow data

All of the linear regressions performed on the data presented in this study had an $R^2$ value between 0.99 and 1.00. It was found that a greater calculated resistance value generally correlated with a greater relative resistance (Table 2). The only exceptions to this observed trend among the catheters presented in this study were 0RO and M2M1; despite having lower relative resistances compared to other samples in their respective groups, these two samples had higher calculated resistances compared to the other samples in those groups. VCTD #1 also shows higher calculated resistance than VCTD #3, despite having a lower time elapsed.

# Discussion

Significant differences were found between the individual VCTD units' relative resistances. However, all the experiments with catheters presented in this study were performed using the same VCTD; variations between VCTDs will not have impacted data acquisition. Furthermore, the standard deviation and range for each VCTD are small (Table 2). Some degree of variance between models is inevitable, even with 3D printing. The base relative resistance of every VCTD will be measured and recorded to account for this variance. As such, the differences in data measurement observed in this study will be accounted for in future studies that use VCTDs for data acquisition.

## The use of relative resistance to infer resistance to CSF flow

The values of resistance calculated in this study are similar to those reported by Cheatle et al. [9]. While the catheters presented in this study cannot be directly compared to those presented in the literature, there is strong evidence to suggest that relative resistance is connected to the catheters' resistance. Indeed, the data presented in this study shows that catheters with a higher relative resistance have a lesser flow rate, evidenced by the higher time interval required for a fixed pressure drop. Thus, the VCTD can be used to quantitatively compare changes in resistance to CSF flow by using relative resistance as a proxy measurement. Furthermore, the VCTD could be used to quantitatively analyze decreases in catheter performance (i.e., reductions in CSF flow over a given time interval) caused by increased resistance to CSF flow.

## Preliminary analysis of explanted catheters from patients with hydrocephalus

An analysis of the samples at the WSU biorepository found that a majority (61.5%) of the failed shunts contained obstructive tissue aggregates [6]. Several explanted catheters presented in this study show significant differences in relative resistances (Fig 4). The prevalence of tissue aggregates and the noticeable increase in relative resistance compared to unused catheters (Tables 2 and 3) strongly suggest a significant relationship between obstruction by tissue aggregates and increased resistance to CSF flow. However, the physiological significance of the biobank data is still unclear. Future work using the VCTD will perform direct comparisons of the relative resistances of different catheter models before and after implantation to quantify how catheter performance changes as resistance changes.

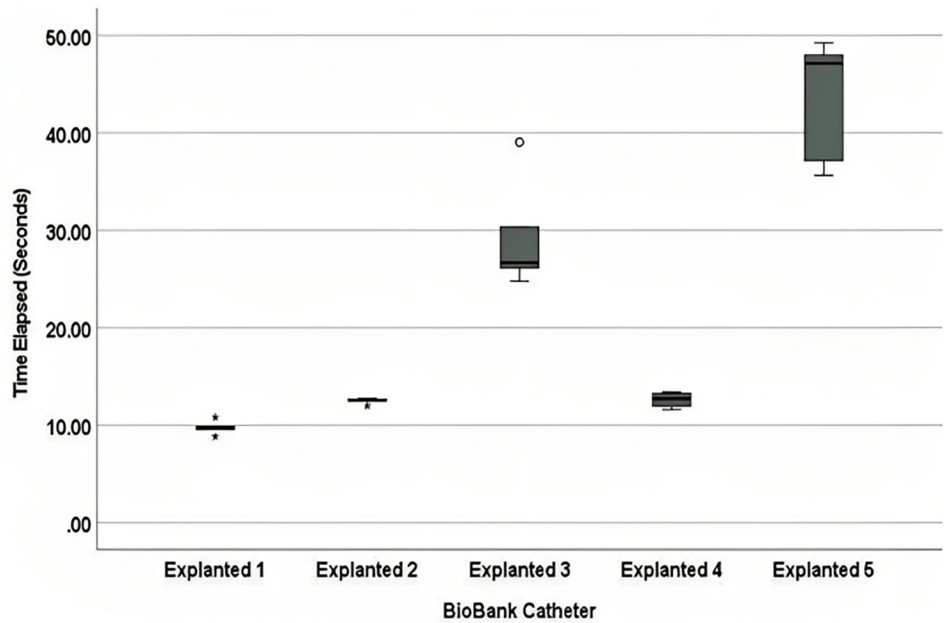

**Fig 4. Explanted catheters data plot.** Box-and-whiskers plot of the time elapsed data for the explanted biobank catheters (n = 5). Asterisks and circles represent data points that are outliers.

## Effects of catheter geometry on resistance to flow

The post-hoc analysis of data from the artificially obstructed catheter suggests that the effect of obstruction on resistance to fluid flow is nonlinear. A recent study used an *in vitro* ventricle model to examine the effect of obstruction on CSF flow; their results indicated that partial obstruction of a catheter does not necessarily reduce CSF flow. Catheters that were close to complete occlusion were still capable of significant CSF outflow [10]. Additionally, the literature suggests that the geometry of the catheters themselves impacts shunt obstruction [3, 11–13]. By testing catheters of varying dimensions and hole geometries, the VCTD can be used to determine whether this nonlinear relationship holds for catheters of different physical designs. If the relationships are different, the VCTD could also be used to help define those relationships.

## Differences in relative resistance among commercial models

The data acquired in this study indicate a significant difference in relative resistances among catheter manufacturers and models. However, the clinical significance of these differences is unclear. There is limited evidence in the literature to suggest any relationship between clinical outcomes and the model of catheter used. The relative resistances between most commercial catheters tested

**Table 3. Time elapsed data and resistance calculations for all groups (with standard deviations) for biobank catheters.**

| Identifier | N | Mean Time Elapsed ± Std. Dev. (s) | Time Elapsed Range (s) | Resistance (Pa•s/m$^3$, 10$^{10}$) | Resistance Std. Dev. (Pa•s/m$^3$, 10$^9$) |
|---|---|---|---|---|---|
| Explanted 1 | 5 | 9.7481 ± 0.70430 | 1.96 | 2.09 | 2.98 |
| Explanted 2 | 5 | 12.4847 ± 0.30090 | 0.78 | 2.61 | 5.62 |
| Explanted 3 | 5 | 29.3962 ± 5.77315 | 14.3 | 3.05 | 3.35 |
| Explanted 4 | 5 | 12.5871 ± 0.78936 | 1.80 | 1.73 | 2.68 |
| Explanted 5 | 5 | 43.4234 ± 6.48526 | 13.6 | 3.98 | 3.26 |

in this study were not significantly different. The study on commercial catheters performed by Galarza et al. found that flow distribution followed similar patterns across the commercial catheters they tested. Their results suggested that hydrocephalus catheters might fail with some degree of uniformity [4]. Future work using the VCTD will determine whether there is a significant variation between the relative resistances of individual catheters from the same model.

### Potential improvements in the VCTD's design

A significant limitation of the VCTD is that it cannot represent the unique conditions that each shunt experiences inside a patient. Measurements taken by the VCTD do not account for differences in the physical dimensions of the various commercial catheter models. Several samples in the WSU biobank underwent electrocautery during the removal process or were fixed for imaging, which may have affected the tissue or the catheter itself. One benefit of the VCTD is that fixing is unnecessary to run experiments; future experiments can be performed on unfixed samples.

Data acquired from the VCTD cannot currently be used to predict whether a shunt at a given resistance will fail. The VCTD is not a bioreactor; it does not simulate flow patterns in the brain. The VCTD is a gravity-driven model, which is not equivalent to how CSF flows through the body. Benchtop pump systems can simulate CSF flow with varying parameters and high precision and have been proposed as modular components of *in vitro* hydrocephalus models [14]. Finally, the artificially obstructed catheter was obstructed using glue, which is notably different from obstruction caused by human tissue. Future experiments with artificially obstructed shunts can use human cell cultures to plug the hole interfaces, providing a more accurate shunt obstruction model.

### Conclusions

This study used a hydrostatic benchtop model to simulate fluid flow through unused and explanted catheters used to treat hydrocephalus. Experiments were also performed on a catheter that had been artificially obstructed and on the benchtop model itself. The results of the experiments showed significant differences in relative resistance between some commercial models and significant differences in relative resistance when an unused catheter's hole interfaces are closed. A non-linear increasing trend in relative resistance was observed as the hole interfaces were progressively closed. Differences in data between individual benchtop models were found to be significantly different. However, these differences can be accounted for in future studies and are unlikely to affect data measurement and analysis meaningfully. This study serves as the foundation for upcoming projects using the VCTD to quantitatively analyze and categorize changes in catheter performance due to changes in resistance to CSF flow.

### Acknowledgments

We would like to thank Prashant Hariharan for his assistance with the WSU biorepository. We would also like to thank Adam Menkara for his assistance in early data collection and conceptualization.

### Author Contributions

**Conceptualization:** Ahmad Faryami, Carolyn A. Harris.

**Data curation:** Pranav Gopalakrishnan, Ahmad Faryami.

**Formal analysis:** Pranav Gopalakrishnan, Ahmad Faryami.

**Funding acquisition:** Carolyn A. Harris.

**Investigation:** Pranav Gopalakrishnan, Ahmad Faryami, Carolyn A. Harris.

**Methodology:** Pranav Gopalakrishnan, Ahmad Faryami, Carolyn A. Harris.

**Project administration:** Ahmad Faryami, Carolyn A. Harris.

**Resources:** Carolyn A. Harris.

**Supervision:** Carolyn A. Harris.

**Visualization:** Pranav Gopalakrishnan, Ahmad Faryami.

**Writing – original draft:** Pranav Gopalakrishnan.

**Writing – review & editing:** Pranav Gopalakrishnan, Ahmad Faryami, Carolyn A. Harris.

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
