## [Decision Letter · Decision Letter 0]

9 Aug 2023

PONE-D-23-10953The development of a portable, benchtop, hydrostatic model to acquire and analyze quantitative data on unused and failed catheters used for the treatment of hydrocephalusPLOS ONE

Dear Dr. Harris,

Thank you for submitting your manuscript to PLOS ONE. After careful consideration, we feel that it has merit but does not fully meet PLOS ONE’s publication criteria as it currently stands. Therefore, we invite you to submit a revised version of the manuscript that addresses the points raised during the review process.

I hope this letter finds you in good health. I wanted to personally express my appreciation for your submission of the manuscript titled "The Development of a Portable, Benchtop, Hydrostatic Model to Acquire and Analyze Quantitative Data on Unused and Failed Catheters Used for the Treatment of Hydrocephalus" to PLOS ONE. I have conducted a comprehensive evaluation of your manuscript, including the insightful feedback provided by our esteemed reviewers, and I am pleased to offer my confidential assessment.

Your manuscript showcases a commendable and innovative effort in designing the Ventricular Catheter Testing Device, a tool specifically developed to evaluate differences in resistance among various catheter models. This endeavor has the potential to make a significant contribution to the field of hydrocephalus treatment. The novel in-vitro model you have introduced holds the promise of opening new avenues for advancing our understanding of intraluminal catheter obstruction mechanisms, potentially reshaping the approach to hydrocephalus treatment. The clarity and organization of your results further substantiate the robustness of your approach and its applicability in the domain of medical device innovation.

The reviewers have commendably recognized your introduction of this innovative in-vitro model, which holds the promise of correlating with clinical patient groups, thereby potentially offering valuable insights into the intricate mechanisms of intraluminal catheter obstruction. The feedback from the reviewers has highlighted areas where further enhancement is possible. They recommend incorporating data that indicates the percentage of double-labeled Vme cells among the labeled cells, a critical step toward achieving a comprehensive understanding of the phenomenon under investigation. Additionally, the suggestion to undertake in-vivo activation experiments to validate the association between the identified double-labeled neurons and the manifested symptoms is well-founded, as it could substantially strengthen the proposed correlation. The reviewers' guidance on refining the materials and methods section signifies a commitment to methodological transparency and scientific rigor. Providing more detailed descriptions of catheter geometry will enable researchers to better grasp the experimental setup and its relevance. Explicitly stating the formula employed for calculating resistance will enhance reproducibility and methodological clarity. Lastly, their recommendation to offer a more precise elucidation of "catheter performance" indicates a strong interest in the practical implications of the research, urging you to clarify how this term aligns with your findings. Collectively, the reviewers' insights provide a comprehensive roadmap for refining and amplifying the impact of this study, aligning well with the journal's commitment to promoting rigorous scientific exploration and application.

I encourage you to thoughtfully address the suggestions and concerns raised by the reviewers, with particular emphasis on enhancing the clarity of certain details in the methodology section. Your consideration of PLOS ONE as a platform for sharing your valuable findings is appreciated. I eagerly anticipate receiving the revised version of your manuscript and continuing the review process.

Thank you for your dedication to advancing scientific knowledge and contributing to the scholarly community.

We look forward to receiving your revised manuscript.

Kind regards,

Naven Jayaprakash

Academic Editor

PLOS ONE

Journal Requirements:

Additional Editor Comments:

I hope this letter finds you in good health. I wanted to personally express my appreciation for your submission of the manuscript titled "The Development of a Portable, Benchtop, Hydrostatic Model to Acquire and Analyze Quantitative Data on Unused and Failed Catheters Used for the Treatment of Hydrocephalus" to PLOS ONE. I have conducted a comprehensive evaluation of your manuscript, including the insightful feedback provided by our esteemed reviewers, and I am pleased to offer my confidential assessment.

Your manuscript showcases a commendable and innovative effort in designing the Ventricular Catheter Testing Device, a tool specifically developed to evaluate differences in resistance among various catheter models. This endeavor has the potential to make a significant contribution to the field of hydrocephalus treatment. The novel in-vitro model you have introduced holds the promise of opening new avenues for advancing our understanding of intraluminal catheter obstruction mechanisms, potentially reshaping the approach to hydrocephalus treatment. The clarity and organization of your results further substantiate the robustness of your approach and its applicability in the domain of medical device innovation.

The reviewers have commendably recognized your introduction of this innovative in-vitro model, which holds the promise of correlating with clinical patient groups, thereby potentially offering valuable insights into the intricate mechanisms of intraluminal catheter obstruction. The feedback from the reviewers has highlighted areas where further enhancement is possible. They recommend incorporating data that indicates the percentage of double-labeled Vme cells among the labeled cells, a critical step toward achieving a comprehensive understanding of the phenomenon under investigation. Additionally, the suggestion to undertake in-vivo activation experiments to validate the association between the identified double-labeled neurons and the manifested symptoms is well-founded, as it could substantially strengthen the proposed correlation. The reviewers' guidance on refining the materials and methods section signifies a commitment to methodological transparency and scientific rigor. Providing more detailed descriptions of catheter geometry will enable researchers to better grasp the experimental setup and its relevance. Explicitly stating the formula employed for calculating resistance will enhance reproducibility and methodological clarity. Lastly, their recommendation to offer a more precise elucidation of "catheter performance" indicates a strong interest in the practical implications of the research, urging you to clarify how this term aligns with your findings. Collectively, the reviewers' insights provide a comprehensive roadmap for refining and amplifying the impact of this study, aligning well with the journal's commitment to promoting rigorous scientific exploration and application.

I encourage you to thoughtfully address the suggestions and concerns raised by the reviewers, with particular emphasis on enhancing the clarity of certain details in the methodology section. Your consideration of PLOS ONE as a platform for sharing your valuable findings is appreciated. I eagerly anticipate receiving the revised version of your manuscript and continuing the review process.

Thank you for your dedication to advancing scientific knowledge and contributing to the scholarly community.

Naveen Jayaprakash

Reviewers' comments:

Reviewer's Responses to Questions

**Comments to the Author**

1. Is the manuscript technically sound, and do the data support the conclusions?

Reviewer #1: Yes

Reviewer #2: Yes

Reviewer #3: Yes

2. Has the statistical analysis been performed appropriately and rigorously? 

Reviewer #1: Yes

Reviewer #2: Yes

Reviewer #3: I Don't Know

3. Have the authors made all data underlying the findings in their manuscript fully available?

Reviewer #1: Yes

Reviewer #2: Yes

Reviewer #3: Yes

4. Is the manuscript presented in an intelligible fashion and written in standard English?

Reviewer #1: Yes

Reviewer #2: Yes

Reviewer #3: Yes

5. Review Comments to the Author

Reviewer #1: This research article describes the analysis of unused and failed hydrocephalus catheters using a novel benchtop hydrostatic model

This is an important area of innovation for both adults and children with the development of novel medical devices such as hydrocephalus catheters as well as the testing equipment to evaluate. Therefore, I am supportive of publications like these and especially to the medical audience who may not be well-exposed to this type of translational research with more emphasis placed on basic science research or translational research

The authors have well described their studies and presented their data in a clear manner that readers can follow. This knowledge will be helpful to future innovators as they develop new hydrocephalus catheters as well as new detection devices for catheter obstruction

One suggested change:

The placement of the figure legends within the text instead of at the end near the figures made this more difficult than usual to review this manuscript

Reviewer #2: This study focuses on investigating the mechanisms of catheter obstruction, the understanding of which is limited. The starting hypothesis is that resistance to fluid flow plays an important role in clogging mechanisms. A gravity-driven device that measures flow in ventricular catheters was developed and evaluated. For this purpose, the resistance of ventricular catheters not used in the treatment of hydrocephalus and failed catheters from a biorepository were quantitatively analyzed. Catheters from three manufacturers were also tested and six models were evaluated in a laboratory model recording time, flow rate and pressure data. Experiments were conducted to evaluate changes in the relative resistance of a catheter as its holes became progressively clogged. Experimental results showed significant differences between the relative resistances of different catheter models just after unpacking. Catheters with artificially plugged holes showed a nonlinear increase in resistance. This novel in vitro model can rapidly correlate with clinical cohorts to identify mechanisms of shunt obstruction in patients. The study advances the understanding of catheter obstruction and its impact on the treatment of hydrocephalus.

In my opinion, the purpose of the paper is attractive and relevant in the context of the subject matter. Achieving a correlation between catheter resistance and catheter performance would be a significant advance in the design of ventricular catheters and I think this can be a first step. However, to give a positive recommendation in the journal PLOS ONE, I suggest improving the clarity and description of some details concerning mainly the materials and methods section. Below, I list some specific points that I believe can be addressed with greater detail or clarity in the paper:

1. It would be interesting to have a better description of the geometric and design characteristics (and/or images) of the catheters tested. Especially those catheters of the group "unused ventricular catheters" and "patient-explanted catheters".

2. Although it is true that the relationship between pressure, caudal and resistance is clear and references are cited in this regard, I believe that the law used and the formula used in the calculation of the resistance shown in Table 1 should be explicitly stated in the methods section.

3. I also think it would be useful to explain what is meant here by "catheter performance": does higher performance mean longer catheter life or does it refer to another type of performance?

Reviewer #3: The new laboratory study coming from the WSU Hydrocephalus team is proposing a novel in-vitro model for use it to examine data on differences in resistance between different catheter models. After these experiments, they might be able to correlate clinical patient groups to identify mechanisms of intraluminal catheter obstruction.

The model referred to as the Ventricular Catheter Testing Device is a simple in-and-out system with a position open for an in-line shunt catheter and data were automatically obtained from tailored specific pressure sensors and flow sensors.

Previously, other hydrocephalus research group, by using CFD analysis, have found that flow distribution followed similar patterns across the different commercial catheters. Their results suggested that hydrocephalus catheters might fail with some degree of uniformity. In fact this is rather true, given that most if not all commercial ventricular catheters follow a similar hole flow configuration.

The authors state that the novel testing device will determine whether there is a variation between the relative resistances of individual catheters from the same model. In fact, this study serves as the base for new projects using the novel testing device to quantitatively analyze and categorize changes in catheter performance due to changes in resistance to CSF flow. These findings can give support to clinical and laboratory studies pertaining to hydrocephalic patients.

The study is well designed and well-written. The main title is overly long. The authors have a solid reputation in the field, and recently published in the same journal another laboratory study: Testing and validation of reciprocating positive displacement pump for benchtop pulsating flow model of cerebrospinal fluid production and other physiologic systems; which belongs to the same field.

Personally, I would shorten the paper, the title, and focus in the testing device itself. Regarding the testing of commercial catheters, I´m keen to know which were the two catheter models that showed significantly differences between time elapsed data among them.

6. PLOS authors have the option to publish the peer review history of their article (what does this mean?). If published, this will include your full peer review and any attached files.

Reviewer #1: No

Reviewer #2: No

Reviewer #3: **Yes: **MARCELO GALARZA, MD, PhD

---

## [Author Response · Author response to Decision Letter 0]

11 Oct 2023

Reviewer #1

1. The placement of figure legends within the text instead of at the end near the figures made it more difficult than usual to review the manuscript.

Thank you for your comment and overall review. The figure legends/captions are placed within the text directly after the paragraph where the respective figures are first cited to match the submission guidelines and formatting requirements of PLOS ONE. If requested, we would be willing to provide an additional, alternative copy of the revised manuscript with the figure captions moved to the end of the document to better facilitate the review process.

Reviewer #2

1. It would be interesting to have a better description… of the catheters tested.

Thank you for your comments and overall review. We have added a table (Table 1) to the Experimental Hypotheses and Methods subsection of the Materials and Methods listing all the identifiers of the samples presented in this study alongside some basic data about the characteristics of the presented catheters: number of holes per row, number of rows, inner diameter, and outer diameter.

2. I believe the law used and the formula used in the calculation of the resistance… should be explicitly stated in the methods section.

We have added two equations (Eq 1 and Eq 2) to the Data Curation and Analysis subsection of the Materials and Methods section. Eq 1 provides the formula relating pressure, resistance, and flow. Eq 2 provides the formula used for the dimensional analysis calculation from of hPa•min/µL to Pa•s/m3.

3. It would be useful to explain what is meant here by “catheter performance.”

We have briefly clarified on what we mean by catheter performance for the purposes of this study in Lines 237-238, which is where the term “catheter performance” is first mentioned in the manuscript.

Reviewer #3

1. I would shorten the paper, the title, and focus on the testing device itself.

Thank you for your comments and overall review. We have shortened both the full title and short title. Unfortunately, we were unable to shorten the manuscript itself, as we did not want to remove context and discussion that we felt were important to the study.

2. I’m keen to know which were the two catheter models that showed significant differences between time elapsed data among them.

We greatly appreciate your interest in this matter! However, we cannot provide that information at this time.

---

## [Decision Letter · Decision Letter 1]

10 Nov 2023

The development of a portable, benchtop, hydrostatic model to acquire and analyze quantitative data on unused and failed catheters used for the treatment of hydrocephalus

PONE-D-23-10953R1

Dear Dr. Harris,

We’re pleased to inform you that your manuscript has been judged scientifically suitable for publication and will be formally accepted for publication once it meets all outstanding technical requirements.

Kind regards,

Naveen Jayaprakash

Academic Editor

PLOS ONE

Additional Editor Comments (optional):

Reviewers' comments:

Reviewer's Responses to Questions

**Comments to the Author**

1. If the authors have adequately addressed your comments raised in a previous round of review and you feel that this manuscript is now acceptable for publication, you may indicate that here to bypass the “Comments to the Author” section, enter your conflict of interest statement in the “Confidential to Editor” section, and submit your "Accept" recommendation.

Reviewer #2: All comments have been addressed

2. Is the manuscript technically sound, and do the data support the conclusions?

Reviewer #2: Yes

3. Has the statistical analysis been performed appropriately and rigorously? 

Reviewer #2: Yes

4. Have the authors made all data underlying the findings in their manuscript fully available?

Reviewer #2: Yes

5. Is the manuscript presented in an intelligible fashion and written in standard English?

Reviewer #2: Yes

6. Review Comments to the Author

Reviewer #2: The authors' revision has improved the paper by providing additional information on certain details that I consider important and that were not present in the first version. As noted in my initial review, the purpose of the article is appealing and relevant in the context of the subject matter, and makes a significant and forward-looking contribution to the understanding of catheter blockage and its impact in the treatment of of hydrocephalus.I therefore recommend its publication in the journal PLOS ONE.

7. PLOS authors have the option to publish the peer review history of their article (what does this mean?). If published, this will include your full peer review and any attached files.

Reviewer #2: No

---

## [Editor Report · Acceptance letter]

16 Nov 2023

PONE-D-23-10953R1 

A novel, benchtop model for quantitative analysis of resistance in ventricular catheters 

Dear Dr. Harris:

I'm pleased to inform you that your manuscript has been deemed suitable for publication in PLOS ONE. Congratulations! Your manuscript is now with our production department. 

Kind regards, 

on behalf of

Dr. Naveen Jayaprakash 

Academic Editor

PLOS ONE